# OpenReview forum: "On the Risk of Misinformation Pollution with Large Language Models"
_EMNLP/2023/Conference — EMNLP 2023 Findings_

### Official Review · Reviewer_k9JU · 2023-08-03

**Soundness:** 4

**Excitement:**

4: Strong: This paper deepens the understanding of some phenomenon or lowers the barriers to an existing research direction.

**Paper Topic And Main Contributions:**

This paper discusses the potential risks of misinformation pollution with LLMs and its impact on ODQA systems. The paper establishes a threat model to assess the extent to which LLMs can be utilized to produce disinformation. It also proposes three defense strategies to mitigate the negative impacts of misinformation pollution.


**Questions For The Authors:**

How do you decide on the number of fake texts injected to the clean corpora for each setting?

**Reasons To Accept:**

1. The paper addresses an interesting and important issue of potential risks associated with LLMs and their impact on information-intensive applications.
2. The paper proposes a threat model which is intuitive and straightforward. The threat model provides notable misleading results in ODQA.
3. The paper presents preliminary experimental results to evaluate the effectiveness of the proposed defense strategies, which can help guide future research and development efforts.

**Reasons To Reject:**

The paper only focuses on one specific type of ODQA systems (retriever-reader). However, there lacks analysis on other types of ODQA systems e.g. retriever-reader.

**Reproducibility:**

2: Would be hard pressed to reproduce the results. The contribution depends on data that are simply not available outside the author's institution or consortium; not enough details are provided.

**Reviewer Confidence:**

4: Quite sure. I tried to check the important points carefully. It's unlikely, though conceivable, that I missed something that should affect my ratings.

---

> ### Author Rebuttal · Authors · 2023-08-28
>
> We thank the reviewer's valuable insights behind the comments and also want to clarify a few points.
>
> > *The paper only focuses on one specific type of ODQA systems (retriever-reader). However, there lacks analysis on other types of ODQA systems e.g. retriever-reader.*
>
> We appreciate the reviewer's feedback. We would like to clarify that the retrieve-and-read is the dominant architecture for existing ODQA systems, following the advent of the first neural QDQA system DrQA (https://arxiv.org/pdf/1704.00051.pdf). Although they all follow the general pipeline of retrieve-and-read, the specific implementations for retriever and reader make them different models. In our work, we have investigated a variety of mainstream architectures, including two main types of retrievers (sparse vs. dense) and the two major types of readers (extractive reader vs. generative reader) in our paper (please see Section 4.2, line 308 to line 338 for details).
>
> The reviewer mentioned we "lack analysis on other types of ODQA systems e.g. retriever-reader." However, we are actually using the retriever-reader architecture. We guess this might be a typo and the reviewer actually means "retriever-generator"? If this is the case, we want to clarify that we have used GPT-3.5 (Lines 327 - 338) as the reader, which represents the retriever-generator architecture.
>
> > *How do you decide on the number of fake texts injected to the clean corpora for each setting?*
>
> Thank you for your insightful question regarding the determination of the number of fake texts injected into the clean corpora for each setting. **There are two factors, generating and processing, determining the final numbers of passages.**
>
> The decision to generate what amount of documents was based on a combination of factors, including the desired level of noise, the distribution of these fake texts in relation to the original data, and the scale of the experiment. To elaborate further, we conducted research reviews and preliminary experiments to identify a reasonable proportion of synthetic text in the ODQA corpora in order to simulate real-world misinformation pollution levels. We determined to generate one fake document for each question for each setting, as the resulting percentage of fake documents in the original corpora would be small (~0.01%), aligning with the estimated percentage of feasible corruption in online data made in this work https://arxiv.org/abs/2302.10149.
>
> The specific passages injected into corpora are processed from the generated documents following standard practices in ODQA. Since the passages were split by 100 words in our case, the generated document lengths determine the amount of injected passages, which we showed in Table 1. We did not further normalize the generated text lengths for two reasons: 1) It would be hard to keep the generated features (word choices, semantics) intact, and 2) We found that usually only one passage in the generated document is retrieved, regardless of document lengths (how many passages one document forms). Therefore, we decided to keep all passages and report the numbers in our paper.

---

### Official Review · Reviewer_UJMA · 2023-08-04

**Soundness:** 3

**Excitement:**

3: Ambivalent: It has merits (e.g., it reports state-of-the-art results, the idea is nice), but there are key weaknesses (e.g., it describes incremental work), and it can significantly benefit from another round of revision. However, I won't object to accepting it if my co-reviewers champion it.

**Paper Topic And Main Contributions:**

The paper investigates if large language models (LLMs) can be deployed to generate misinformation. In specific, the authors consider the task of open domain question answering (ODQA) and utilize the generative capability of LLMs to generate noisy documents that can mislead an ODQA model to come up with the wrong answer. Experiments on two real world datasets reveal that introducing these noisy documents can negatively influence the performance of the models. Additionally, the paper also comes up with some simple defense strategies to combat such misinformation.

**Questions For The Authors:**

Watermarking is also being deployed, whereby the model leaves a mark on the generated text which can then be utilized to distinguish it from a human generated text. Do you think such information could be utilized to build defenses?

How easy or difficult it is for humans to identify a document generated by your method? Did you perform any study with regard to this?

**Reasons To Accept:**

Deals with an interesting and timely problem. The paper is also nicely written and easy to follow.

Experiments are performed on real-world datasets.

Proposes defense strategies to combat misinformation.



**Reasons To Reject:**

Given the ever improving generative and instruction following capabilities of LLMs, it is intuitive that they come up with such documents supporting misinformation. Hence, I am a bit unconvinced about the novelty of this paper. The prompts also seem to be straightforward. That LLMs are good at generating authoritative text is already well established.

I would imagine that in addition to documents with misinformation, there will be a lot of supporting documents containing the true information. It could be that one needs to inject a lot more noisy documents for the systems to generate the wrong answer. From table 1, it looks like a small fraction is enough to worsen the performance of the ODQA model. Maybe it is worth investigating what happens when you continue to add more and more noisy documents. Further investigations are required.

**Reproducibility:**

4: Could mostly reproduce the results, but there may be some variation because of sample variance or minor variations in their interpretation of the protocol or method.

**Reviewer Confidence:**

4: Quite sure. I tried to check the important points carefully. It's unlikely, though conceivable, that I missed something that should affect my ratings.

---

> ### Author Rebuttal · Authors · 2023-08-28
>
> We thank the reviewer's valuable insights behind the comments and also want to clarify a few points.
>
> > *Given the ever-improving generative and instruction-following capabilities of LLMs, it is intuitive that they come up with such documents supporting misinformation. Hence, I am a bit unconvinced about the novelty of this paper.*
>
> While it's true that the generative abilities of LLMs are well-known, we argue that the community has not yet thoroughly explored the **implications** of these capabilities, especially whether the generative abilities can be misused to produce misinformation and pollute the corpus used for knowledge-intensive applications such as QA. Our work highlights potential vulnerabilities in ODQA systems when exposed to both unintentional and deliberate misinformation. We also offer preliminary attempts to address this issue, laying the groundwork for more in-depth follow-up research. Therefore, our contribution is not merely an investigation into the misinformation-generating capacities of LLMs, but more on a comprehensive study of **the implication of misinformation pollution and the potential countermeasures.**
>
> > *The prompts also seem to be straightforward. That LLMs are good at generating authoritative text is already well established.*
>
> As suggested by the paper title, the motivation of our study is to uncover **the risk of misinformation pollution** amplified by the widely use of LLMs. Our choice of using "straightforward" prompts was intentional, aiming to mimic real-world interactions between users or potential attackers and LLMs, as detailed in Lines 193-204 of our paper. The fact that even these simple prompts can lead to the generation of convincing misinformation accentuates the urgency of this issue. The attackers do not need advanced prompting skills to exploit LLMs for misinformation creation, posing further risks to downstream applications. This reiterates not only the likelihood but also the significance of information pollution, emphasizing its necessity as a topic for serious attention.
>
> > *I would imagine that in addition to documents with misinformation, there will be a lot of supporting documents containing the true information. It could be that one needs to inject a lot more noisy documents for the systems to generate the wrong answer. From Table 1, it looks like a small fraction is enough to worsen the performance of the ODQA model. Maybe it is worth investigating what happens when you continue to add more and more noisy documents. Further investigations are required.*
>
> We thank the reviewer for raising this interesting point on the potential presence of both supporting and contradictory documents in the input data. We agree that the extent of the performance degradation could depend on the input composition. The following table shows the QA performance (DPR+FiD ODQA system) on the NQ-1500 dataset under the CtrlGen setting, with different numbers of noisy documents generated per question and the different number of documents utilized by the reader.
>
> |              | 1 noisy docs | 3 noisy docs |
> | ------------ | ------------ | ------------ |
> | 100 contexts | 42.27        | 31.73        |
> | 10 contexts  | 36.93        | 29.20        |
>
> **Our preliminary experiments showed that the larger the portion of misinformation in the retrieved documents, the worse the ODQA performance in general.** We will add the above additional results into the camera-ready version and further conduct a more large-scale analysis of how varying levels of noise in input data might influence the prevalence of misinformation.
>
> > *Watermarking is also being deployed, whereby the model leaves a mark on the generated text which can then be utilized to distinguish it from a human-generated text. Do you think such information could be utilized to build defenses?*
>
> We concur with the reviewer that watermarking is an emerging and promising technique to distinguish AI-generated texts. Unfortunately, the first watermarking work for LLM (https://arxiv.org/abs/2301.10226) was published after the submission of this paper, so we haven't had a chance to explore it. Nonetheless, we believe watermarking could potentially provide an additional signal to distinguish model-generated text from human-generated text, contributing to potential defense strategies against the misuse of generated content. Compared to neural network classifiers we experimented with in section 5, we believe watermarked machine-generated text would be more successfully and reliably detected, according to some recent works https://arXiv.org/abs/2305.08883. However, currently, we must know the type of watermark beforehand to detect watermarked text, making it impractical at this moment (various LLMs and no unified watermarking in practice). This practicality problem is similar to the problem of neural network classifiers, which we discussed in Lines 517 - 522. We appreciate the reviewer's insightful suggestion and we aim to explore watermarking as a potential defense technique in the camera-ready version, if time permits.
>
> > *How easy or difficult it is for humans to identify a document generated by your method? Did you perform any study with regard to this?*
>
> As for the ease of human identification of documents generated by our method, we conducted a small-scale human study to explore the detectability of misinformation. We randomly sampled 50 fake documents generated under CtrlGen and 50 corresponding most relevant Wikipedia passages. We employed experimental settings described in Clark et al. (https://aclanthology.org/2021.acl-long.565.pdf,), where participants need to rate each text on a 4-point scale. We hired three college students to each annotate the 100 documents. Similar to their findings, we found humans cannot reliably differentiate machine-generated misinformation from their Wikipedia counterparts, with an average overall accuracy of 57%. Note that we observed great performance improvements in the second half of experiments for all participants, which could mean that 57% is an overestimation of human capabilities since the participants displayed signs of learning the sampled data in our experiments. We will include this experiment in the camera-ready version. We thank the reviewer for pointing this out.

---

### Official Review · Reviewer_Qkxs · 2023-08-04

**Soundness:** 3

**Excitement:**

3: Ambivalent: It has merits (e.g., it reports state-of-the-art results, the idea is nice), but there are key weaknesses (e.g., it describes incremental work), and it can significantly benefit from another round of revision. However, I won't object to accepting it if my co-reviewers champion it.

**Paper Topic And Main Contributions:**

The paper investigates how generative language models can be used to generate credible misinformation. Two main scenarios have been addressed, i.e. intentional and unintentional, about misinformation creation. The authors experimentally evaluated the risk of fake information creation and proposed three mitigation strategies named misinformation detection, vigilant prompting, and reader ensemble.
The obtained results confirmed what was previously demonstrated in similar papers, highlighting however that a few strategies can mitigate in a very good way misinformation creation.


**Reasons To Accept:**

The main reasons to accept are:
- fair comparison of existing pre-trained generative language models
- the mitigation strategies demonstrate to effectively contribute to limit the diffusion of misinformation generated by LLM


**Reasons To Reject:**

The main reasons to reject are listed as follows:

1) The mitigation policy based on QA prompts is not detailed enough. How the prompts have been created?

2) The equation reported at line 500 is not formally correct. Probably, if I've understood well, a "for any value of j" is missing. What is the effective meaning of the indicator function? Does it measure how many times the answers of passages are equal?
It is then not clear how the "voted answer" is used to minimize the misinformation.

3) The three research questions listed in the introduction are not properly addressed. In particular:
- "(1) To what extent can modern LLMs be utilized for generating credible-sounding misinformation?" is a very general question that has not been specifically tackled in the paper. The authors demonstrated that generative models can be easily used to create misinformation (showing different generation settings) but not to what extent.
- (2) "What potential harms can arise from the dissemination of neural-generated misinformation in information-intensive applications, such as information retrieval and question-answering? " Table 7, which is in the appendix, partially gives an answer to the question showing which scenario is more problematic than others but there is no indication about What potential harms can arise from the dissemination. Table 7 basically says that retrieval is affected by the generated language model text, but there aren't any other indications about any other potentially affected task.

**Reproducibility:**

2: Would be hard pressed to reproduce the results. The contribution depends on data that are simply not available outside the author's institution or consortium; not enough details are provided.

**Reviewer Confidence:**

4: Quite sure. I tried to check the important points carefully. It's unlikely, though conceivable, that I missed something that should affect my ratings.

---

> ### Author Rebuttal · Authors · 2023-08-28
>
> We appreciate the valuable insights behind the comments and also want to clarify a few points.
>
> > *The mitigation policy based on QA prompts is not detailed enough. How the prompts have been created?*
>
> We appreciate the reviewer's feedback and include a more detailed description of our QA prompt creation process as follows. We drew inspiration from an empirical prompt engineering guide [Adversarial Prompting | Prompt Engineering Guide (promptingguide.ai)](https://www.promptingguide.ai/risks/adversarial#defense-tactics). It employs prompting to effectively handle a malicious prompt injection attack, which shares similar traits with our attack settings. Then, we researched relevant papers and based our prompt templates on https://arxiv.org/abs/2305.01579 and https://arxiv.org/abs/2306.10702, with manual revisions to make them suit our needs.
>
> > *The equation reported at line 500 is not formally correct. Probably, if I've understood well, a "for any value of j" is missing. What is the effective meaning of the indicator function? Does it measure how many times the answers of passages are equal? It is then not clear how the "voted answer" is used to minimize the misinformation.*
>
> Thank you for pointing out the error in our equation. The correct version is the one below, with the `argmax` argument changed from `a_i` to `a_j` . It does, measure how many times the $k$ predictions are equal, with the $k$ predictions made based on different partitions of retrieved documents.
>
> $$ a = \underset{a_j}{\operatorname{argmax}}\left(\sum_{i=1}^{k}\operatorname{I}(a_i=a_j)\right) $$
>
> **The mechanism behind the "voted answer" is to reduce the influence of any single piece of retrieved evidence while maintaining the machine reader's performance.** Traditionally, machine readers are trained to focus on evidence with a high probability of containing answers in ODQA. This works well when the corpora are thoroughly filtered, but it also makes them vulnerable to unfiltered misinformation. In light of Lewis et al.’s [arxiv.org/pdf/2005.11401.pdf](https://arxiv.org/pdf/2005.11401.pdf) approach that conditioned token prediction on different documents, we experimented with multiple ideas on making the answer prediction less reliant on any single piece of evidence, including re-training, data augmentation, and the voting technique. Ultimately, we found the voting technique is a simple but effective method.
>
> > *To what extent can modern LLMs be utilized for generating credible-sounding misinformation?" is a very general question that has not been specifically tackled in the paper. The authors demonstrated that generative models can be easily used to create misinformation (showing different generation settings) but not to what extent.*
>
> We agree with the reviewer that the first research question, "To what extent can modern LLMs be utilized for generating credible-sounding misinformation?" is a general research question. It describes the boarder scope of this paper. More precisely, our paper aims to investigate the feasibility of using LLMs to generate misinformation under various generation settings that mimic plausible real-world scenarios. Additionally, we qualitatively assess the impact of such misinformation on downstream NLP tasks like QA. To better align with our focus, we will rephrase the research question in the introduction into: "What is the feasibility of using LLMs to generate misinformation harmful to downstream NLP applications?" Thanks for pointing this out.
>
> > *"What potential harms can arise from the dissemination of neural-generated misinformation in information-intensive applications, such as information retrieval and question-answering? " Table 7, which is in the appendix, partially gives an answer to the question showing which scenario is more problematic than others but there is no indication about What potential harms can arise from the dissemination. Table 7 basically says that retrieval is affected by the generated language model text, but there aren't any other indications about any other potentially affected task.*
>
> We acknowledge that "What potential harms can arise from the dissemination of neural-generated misinformation in information-intensive applications?" is a general research problem and beyond the scope of a single paper. Therefore, in our paper, we primarily focus on investigating the harm of disseminating neural-generated misinformation to open-domain question-answering, a representative information-intensive application. Our work primarily demonstrated the brittleness of ODQA systems under practical misinformation pollution attacks (in Table 2, and line 421 to line 442), which could be exploited by malicious actors to manipulate the information presented to users. We will add a clarification in the camera-ready version to better reflect our focus. Thanks for pointing this out.
>
> Regarding the other potentially affected factors, we have shown that retrieval is affected by the generated misinformation. The change in retrieval results can not only harm the ODQA performance, but also it can subsequently influence humans. For instance, the 'features snippets' provided by search engines, which attempt to answer user queries directly by extracting from relevant search results (making them essentially an ODQA service), have been found convincing and influential to humans (https://dl.acm.org/doi/abs/10.1145/3498366.3505766) and have large-scale undesired consequences (https://dl.acm.org/doi/abs/10.1145/3576840.3578323). Since we focus on how neural misinformation can affect ODQA systems, discussing other potentially affected parties (e.g., humans) is beyond the scope. Nonetheless, we will add the above discussions to highlight the boarder implications of the dissemination of neural-generated misinformation in information-intensive applications.

---

### Meta-Review · Area_Chair_3NKB · 2023-09-30

**Recommendation:** 3

**Metareview:**

This paper explore how Large Language Models (LLMs) can be misused to generate misleading or false information that can affect Open-Domain Question Answering (ODQA) systems. Simulate different scenarios of misinformation generation and measure the impact on ODQA performance. Propose some defense strategies to detect and prevent misinformation from LLMs. Call for more research and collaboration to address the challenges of LLM-generated misinformation and to ensure the ethical use of these models.

---

### Decision · Program_Chairs · 2023-10-07

**Decision:**

Accept-Findings

**Comment:**

This paper explore how Large Language Models (LLMs) can be misused to generate misleading or false information that can affect Open-Domain Question Answering (ODQA) systems. Simulate different scenarios of misinformation generation and measure the impact on ODQA performance. Propose some defense strategies to detect and prevent misinformation from LLMs. Call for more research and collaboration to address the challenges of LLM-generated misinformation and to ensure the ethical use of these models.